# Pharmacological and Genetic Disruption of C-Type Natriuretic Peptide (*nppcl*) Expression in Zebrafish (*Danio rerio*) Causes Stunted Growth during Development

**DOI:** 10.3390/ijms241612921

**Published:** 2023-08-18

**Authors:** Andrew J. Lessey, Samantha M. Mirczuk, Annisa N. Chand, Deborah M. Kurrasch, Márta Korbonits, Stijn J. M. Niessen, Craig A. McArdle, Imelda M. McGonnell, Robert C. Fowkes

**Affiliations:** 1Endocrine Signalling Group, Royal Veterinary College, University of London, Royal College Street, London NW1 0TU, UK; alessey@antlerbio.com (A.J.L.); samantha.byers@admin.cam.ac.uk (S.M.M.); a.n.chand@gmail.com (A.N.C.); 2Comparative Biomedical Sciences, Royal Veterinary College, University of London, Royal College Street, London NW1 0TU, UK; 3Department of Medical Genetics, University of Calgary, Calgary, AB T2N 4N2, Canada; kurrasch@ucalgary.edu; 4Centre for Endocrinology, William Harvey Research Institute, Barts and the London School of Medicine and Dentistry, Queen Mary University of London, London EC1M 6BQ, UK; m.korbonits@qmul.ac.uk; 5Clinical Sciences & Services, Royal Veterinary College, Hawkshead Lane, North Mymms, Hatfield, Hertfordshire AL9 7TA, UK; sniessen@rvc.ac.uk; 6Veterinary Specialist Consultations, Loosdrechtseweg 56, 1215 JX Hilversum, The Netherlands; 7Department of Translational Science, Bristol Medical School, University of Bristol, Whitson Street, Bristol BS1 3NY, UK; craig.mcardle@bristol.ac.uk; 8Endocrine Signaling Group, Department of Small Animal Clinical Sciences, College of Veterinary Medicine, Michigan State University, Wilson Road, East Lansing, MI 48824, USA

**Keywords:** zebrafish, natriuretic peptides, growth

## Abstract

Human patients with mutations within *NPPC* or *NPR2* genes (encoding C-type natriuretic peptide (CNP) and guanylyl cyclase-B (GC-B), respectively) display clinical signs associated with skeletal abnormalities, such as overgrowth or short stature. Mice with induced models of *Nppc* or *Npr2* deletion display profound achondroplasia, dwarfism and early death. Recent pharmacological therapies to treat short stature are utilizing long-acting CNP analogues, but the effects of manipulating CNP expression during development remain unknown. Here, we use *Danio rerio* (zebrafish) as a model for vertebrate development, employing both pharmacological and reverse genetics approaches to alter expression of genes encoding CNP in zebrafish. Four orthologues of CNP were identified in zebrafish, and spatiotemporal expression profiling confirmed their presence during development. Bioinformatic analyses suggested that *nppcl* is the most likely the orthologue of mammalian CNP. Exogenous CNP treatment of developing zebrafish embryos resulted in impaired growth characteristics, such as body length, head width and eye diameter. This reduced growth was potentially caused by increased apoptosis following CNP treatment. Expression of endogenous *nppcl* was downregulated in these CNP-treated embryos, suggesting that negative feedback of the CNP system might influence growth during development. CRISPR knock-down of endogenous *nppcl* in developing zebrafish embryos also resulted in impaired growth characteristics. Collectively, these data suggest that CNP in zebrafish is crucial for normal embryonic development, specifically with regard to growth.

## 1. Introduction

C-type natriuretic peptide (CNP) is the third member of the natriuretic peptide family that also includes atrial- and B-type natriuretic peptides (ANP and BNP, respectively) [1]. Whilst ANP and BNP are predominantly viewed as cardiovascular peptide hormones, both having profound effects on cardiac output, blood pressure and natriuresis, CNP exhibits less natriuretic activity and instead acts as an autocrine/paracrine regulator in many peripheral tissues, including the endocrine system and endothelium [2,3]. However, CNP (*Nppc*) is the predominant natriuretic peptide of the central nervous system, where it controls endocrine, neuronal and glial function [2,4,5]. In mice, CNP has been shown to control axonal development and bifurcation [6,7,8].

The vast majority of natriuretic peptide effects are mediated by the particulate guanylyl cyclase receptors, specifically GC-A (*Npr1*) and GC-B (*Npr2*), leading to elevated cGMP concentrations [1,4]. A third natriuretic peptide receptor, the clearance receptor (*Nprc*) binds all three natriuretic peptides with equal affinity, but does not possess intrinsic guanylyl cyclase activity [9], and acts to manipulate local concentrations of natriuretic peptides [1,4]. However, *Nprc* does couple to signaling pathways in some tissues, by virtue of interactions with G-proteins [9]. Of the three main natriuretic peptides, CNP is the most conserved between species [2]. Transgenic mouse models of disrupted *Nppc* or *Npr2* expression are phenotypically similar, and exhibit profound growth abnormalities; specifically, these mice exhibit achondroplasia due to impaired endochondral ossification, leading to pronounced dwarfism [10,11,12] with early death in 90% of these cohorts. Interestingly, this increased mortality can be rescued by transgenic expression of *Nppc* in the growth plate [13]. Several human mutations in either *NPPC* or *NPR2* have been reported, that also result in a clinical phenotype consistent with skeletal abnormalities, leading to short-stature under conditions of downregulation [14,15,16], and tall-stature and skeletal overgrowth in conditions of upregulation [16,17,18,19]. In fact, *NPR2* and *NPPC* are rare examples of genes that confer a bidirectional effect on growth [20]. Pediatric patients presenting with idiopathic short stature have been successfully treated with long-acting CNP analogs, with the compounds Vosoritide and TransCon CNP recently completing clinical trials [21,22,23,24]. However, the effects of CNP treatment during development are yet to be determined.

The piscine natriuretic peptide system has been extensively characterized in several species, including members of the elasmobranch and teleost families [25,26,27]. In zebrafish (*Danio rerio*), orthologues exist for all three mammalian natriuretic peptide genes (*nppa*, *nppb*, *nppc*) as well as ventricular natriuretic peptide. This genetically tractable species is particularly useful for developmental studies, due to the relative ease of delivery in terms of exogenous compounds, as well as through genetic approaches, such as morpholinos and CRISPR [28,29]. Zebrafish are also useful models in which to investigate the endocrine control of growth, as they express many of the same hormones as humans [30]. Expression of natriuretic peptide receptors during development has been described in zebrafish, with *npr2* being expressed throughout early development and strongly in the adult brain [31]. Teleosts are known to express at least four genes encoding CNP [25,32], with gene duplication events giving rise to different orthologues [27,33]. However, functional investigations of CNP in the developing zebrafish embryo has yet to be reported.

Here, we describe the CNP system in developing zebrafish embryos, and identify *nppcl* as the likely zebrafish orthologue of human CNP. Furthermore, pharmacological and genetic manipulation of *nppcl* expression causes significant impairment in growth characteristics during embryonic development, that is partially associated with an increase in apoptosis. 

## 2. Results

### 2.1. Sequence Alignment and mRNA Expression Profiling in Developing Zebrafish Embryos

Using published sequences of whole-natriuretic-peptide precursors (both known and predicted), a multiple sequence alignment was performed using Clustal Omega (http://www.ebi.ac.uk/Tools/msa/clustalo/, accessed on 1 January 2015). All whole-precursor sequences for *Homo sapiens* (Human), *Mus Musculus* (Mouse), *Rattus norvegicus* (Norwegian Rat), *Gallus gallus domesticus* (Chicken), *Danio rerio* (Zebrafish), *Oreochromis niloticus* (Nile Tilapia) and *Oryzias latipes* (Japanese Medaka) were included. The resulting sequence alignment was used to perform evolutionary genetic analysis using MEGA5 (http://megasoftware.net, accessed on 1 January 2015) and a phylogenetic tree created using the maximum-likelihood method and Tamura–Nei substitution model. A search through genetic databases for zebrafish natriuretic peptide precursors uncovered four genes encoding CNP (*nppc*-like, *nppc2*-like, *cnp3*-like and *nppc4*-like), one for ANP (*nppa*) and two for BNP (*nppb* and *bnp*), but no sequence for the teleost specific VNP (see Figure 1A). The resulting sequences were aligned with genetic sequences for natriuretic peptides from higher vertebrates (human, murine and chicken) along with natriuretic peptides from other teleost species that had been previously characterized (Japanese medaka and Nile tilapia). Phylogenetic analyses demonstrated that each zebrafish gene encoding CNP formed a separate clade with its orthologue from other teleost species. Zebrafish CNP-1 (*nppcl/nppc*-like) formed a clade with other known teleost CNP-1 sequences (Figure 1A). Similarly, CNP-3 (*cnp3*-like) formed a clade with other known teleost CNP-3 transcripts and Chicken CNP-3. The predicted sequence of zebrafish CNP-2 (*nppc2*-like) was placed on a different branch to other teleost CNP-2 sequences. However, zebrafish CNP-4 (*nppc4*-like) formed a large clade with known teleost CNP-4 sequences and mammalian CNP sequences. Bootstrap analysis values at nodes across the phylogenetic tree were commonly low, primarily due to the highly variable prosegment within each sequence. Collectively, these data suggest that zebrafish possess four distinct CNP isoforms, similar to other teleost species, with CNP-1 and CNP-4 being the most likely orthologues of mammalian CNP.

We performed a similar sequence alignment for natriuretic peptide receptors. Mining through genetic databases for known and predicted zebrafish natriuretic peptide receptor sequences, uncovered one gene encoding GC-B (*nprb*), two encoding GC-A (*npra* and *npr1a*) and one encoding the clearance receptor (*nprc*) (Appendix A). The resulting phylogenetic tree organized the receptors into three distinct groups. NPR1 (GC-A) receptors including zebrafish *npr1a* and *npr1b* were placed within the same clade for most species. The predicted zebrafish *npr2*-like sequence was placed in a clade with all other NPR2 (GC-B) sequences and formed a smaller clade with the other teleost GC-B receptor sequences. NPR3 receptor sequences were placed on a branch independent of GC-A and GC-B receptors, due to their structural difference and lack of a guanylyl cyclase domain.

In order to evaluate the structure of the mature zebrafish CNPs, in silico analyses were performed, initially assessing enzyme cleavage sites within the precursor sequences. Zebrafish amino acid sequences were obtained from the NCBI Protein database (http://www.ncbi.nlm.nih.gov/protein, accessed on 1 January 2015) and potential processing endopeptidase cleavage sites identified. Of the four zebrafish CNP sequences, only CNP-4 possessed a furin-processing signal; the other CNP precursors are predicted to be cleaved at dibasic amino acid motifs. The sequences of the predicted mature peptides were aligned with mature human and murine CNP sequences to identify conserved regions. The resulting alignment highlights the strongly conserved 17 amino acid disulfide ring structure, characteristic of all natriuretic peptides, with distinct regions of high or complete conservation within the ring structure highlighted in grey. Zebrafish CNP-1 (*nppcl*/*nppc*-like), CNP-2 (*nppc2*-like) and CNP-3 (*cnp3*-like) formed mature peptides that are 22 amino acids in length and have the typical absence of a carboxyl terminal extension indicating a high level of structural similarity to their mammalian orthologues. Zebrafish CNP-4 (*nppc4*-like) appeared to be substantially different to the other peptides examined. It encoded a considerably larger peptide by comparison, at 37 amino acids long; the majority of this increased size could be attributed to the highly irregular presence of a carboxyl-terminal extension and a longer amino terminal extension. Sequence homology was examined in comparison with mature human CNP and represented as a percentage for each peptide (Figure 1B). Zebrafish CNP-1 was the most structurally similar to human CNP, with 86% sequence homology, attributable to the identical sequence in their disulfide ring sequences. Zebrafish CNP-4 was the most dissimilar of zebrafish CNPs to human CNP, with 38% sequence homology and obvious structural deviations in comparison. However, CNP-4 was the only zebrafish CNP found to possess a furin-processing signal. Collectively, these data suggest that zebrafish CNP-1 is most structurally similar to mammalian CNP orthologues.

The next aim was to examine the temporal expression of these genes during development, along with other components of the zebrafish natriuretic peptide system, since several studies in mice and humans have reported CNP and its receptor GC-B to be expressed throughout early development in embryonic and fetal tissues [8,34,35]. Total RNA was extracted from pools of Tupfel-long fin (TL) wild-type (Wt) zebrafish embryos (~50) collected at 6, 12, 24, 36, 48, 72, 96 and 120 h post-fertilization (hpf) along with adult zebrafish head (AH) and torso (AT) segments. As shown (Figure 1C), all C-type natriuretic peptide genes were expressed early on within zebrafish development. However, there were noticeable differences in apparent temporal expression between each gene, with CNP-1 (*nppc*-like) and CNP-4 (*nppc4*-like) being expressed at earlier developmental time points than CNP-2 (*nppc2*-like) and CNP-3 (*cnp3*-like). The temporal expression profile of natriuretic peptide receptors (Figure 1C) were more consistent in the developing zebrafish, with all receptors including GC-B (*nprb*) expressed at each developmental time point. The expression of atrial natriuretic peptide (*nppa*) and B-type natriuretic peptide (*nppb*) was also detected by PCR, although transcript detection was negligible for both genes before 24 hpf. 

### 2.2. Spatial Expression of CNP-1 (nppc-like) in Developing Zebrafish Embryos

To determine the spatial expression profile of transcripts encoding the four orthologues of CNP, whole-mount in situ hybridization was performed on 96 hpf wild-type zebrafish embryos. As shown (Figure 2A–F), *nppcl* was expressed throughout the brain, highly concentrated along the dorsal midline in the forebrain and midbrain, with weaker staining in the anterior regions of the hindbrain. Specific expression was also detected in neuroendocrine regions, such as the presumptive pituitary and hypothalamus (Figure 2A). No staining was visible with the sense *nppc-like* probe (Figure 2E,F).

Similar temporal expression profiles were observed when using probes specific for *nppc2*, *cnp3* and *nppc4* (Appendix A), as well as for *nprb* (Appendix A), although *nprb* expression was also detected in ventral regions of the trunk. Collectively, these data suggest that CNP orthologues and the GC-B receptor are expressed predominantly in the developing brain and nervous system in zebrafish embryos.

### 2.3. Quantitative Temporal Expression Profiling of Natriuretic Peptide Components in Developing Zebrafish Embryos

Conventional end point RT-PCR and in situ hybridization showed the expression of multiple CNP orthologues and related receptors in developing zebrafish embryos. To investigate quantitative changes in these transcripts during development, a multiplex RT-qPCR assay was performed to examine natriuretic peptide gene expression in developing zebrafish embryos between 24 and 120 hpf. Specific transcripts for all genes of interest were detected by the multiplex assay (Figure 3A). To compare relative changes in gene expression, transcript detection was displayed as fold change compared to 24 hpf. As shown (Figure 3B), there was a rapid increase in the expression of both *nppa* and *bnp* transcripts by 48 hpf (to 9.14 ± 0.5 fold (**** *p* < 0.0001) and 5.5 ± 0.4 fold (**** *p* < 0.0001) over 24 hpf, for *nppa* and *bnp*, respectively), before stabilizing for the remaining time points. In contrast, the expression of transcripts encoding the four distinct CNP genes showed different temporal profiles. *nppc2* and *nppc4* were expressed at 24 hpf but did not vary significantly throughout the time course studied (Figure 3C). However, *nppc3* gradually increased, with highest expression detected at 96 hpf (4.8 ± 0.2-old (**** *p* < 0.0001), compared to 24 hpf). A similar gradual increase in transcript expression was seen with *nppcl*, yet the magnitude of the fold increase was much greater, rising to 45.7 ± 2.8 fold (**** *p* < 0.0001) compared to 24 hpf (Figure 3D). Transcripts for all three natriuretic peptide receptors were detected at 24hpf, but only *npr1a* expression increased significantly by 120 hpf (to 1.7 ± 0.1 fold, **** *p* < 0.0001) (Figure 3E). Finally, transcripts were detected for *furina*, *furinb* and *corin* at 24 hpf and remained stable. Collectively, these data reveal different temporal expression profiles of the natriuretic peptides, their receptors and processing enzymes, during zebrafish development, the most prominent of which is *nppcl*. 

### 2.4. Effect of Exogenous CNP on Growth in Developing Zebrafish Embryos

Having determined the expression profile of natriuretic peptide components during zebrafish development, potential effects of exogenous CNP on the growth and development of zebrafish embryos were investigated. To confirm that zebrafish respond to CNP, dispersed primary cultures of zebrafish embryos were stimulated with 100 nM of mammalian CNP for up to 180 min, in the presence of 1 mM IBMX to inhibit phosphodiesterase activity. As shown (Figure 4A), total cGMP concentrations modestly increased at 180 min from 1.6 ± 0.04 pmol/mL to 4.7 ± 1.0 pmol/mL (*p* < 0.01), indicating the likely presence of a CNP-sensitive guanylyl cyclase. We then treated zebrafish embryos within 2 hpf with either 0 or 100 nM CNP in aquaria water, for up to 72 h. Fresh treatments were administered each day. As shown (Figure 4B), exposure to exogenous mammalian CNP caused a significant reduction in head width at 48 hpf (from 334.6 ± 7.5 µm to 261.9 ± 6.7 µm, in untreated and CNP-treated embryos, respectively; *p* < 0.0001) and 72 hpf (from 409.5 ± 5.8 µm to 389.3 ± 5.1 µm, in untreated and CNP-treated embryos, respectively; *p* < 0.05). Body length of CNP-treated zebrafish embryos was also significantly reduced compared to untreated embryos at 48 hpf (Figure 4C, from 3045.4 ± 27.6 µm to 2860.8 ± 45.7 µm, in untreated and CNP-treated embryos, respectively; *p* < 0.001) and 72 hpf (from 3524.5 ± 41.9 µm to 3398.7 ± 20.9 µm, in untreated and CNP-treated embryos, respectively; *p* < 0.01). Furthermore, eye diameter was significantly reduced in CNP-treated zebrafish embryos compared to untreated embryos, at each time point investigated (Figure 4D; 24 hpf, 141.4 ± 9.2 µm to 114.0 ± 6.7 µm, *p* < 0.05, in untreated and CNP-treated embryos, respectively; 48 hpf, 182.5 ± 3.1 µm to 160.7 ± 3.9 µm, *p* < 0.001, in untreated and CNP-treated embryos, respectively; 72 hpf, 210.7 ± 2.4 µm to 202.3 ± 2.0 µm, *p* < 0.01, in untreated and CNP-treated embryos, respectively).

To determine a potential mechanism for the observed growth inhibition of zebrafish embryos in the presence of 100 nM CNP, we next performed TUNEL staining on wholemount zebrafish embryos treated with 0 or 100 nM CNP for up to 72 hpf. As shown (Figure 4E,F), 100 nM CNP caused a reduction in TUNEL positive cells of zebrafish embryos after 48 hpf (from 3900.6 ± 634.9 pixels to 1823.5 ± 318.6 pixels, *p* < 0.05, in untreated and CNP-treated embryos, respectively), but a significant increase in TUNEL positive cells after 72 hpf (from 1646.5 ± 242.5 pixels to 4339.0 ± 532.7 pixels, *p* < 0.001, in untreated and CNP-treated embryos, respectively). These data show that exposure to exogenous CNP during development can result in impaired growth characteristics, potentially due to an increase in apoptosis. 

We next examined the effect of exogenous CNP treatment on zebrafish larvae locomotion. Individual larvae that had been incubated with 0 or 100 nM CNP were assessed for 24 h locomotory behavior at 72 hpf. As shown (Figure 4G–K), although CNP-treated zebrafish larvae showed a tendency for increased distance, movement and velocity, none of this activity attained statistical significance, suggesting that CNP does not affect zebrafish locomotion during development. 

### 2.5. Effect of Exogenous CNP on Expression of Natriuretic Peptide Components in Developing Zebrafish Embryos

CNP analogs are currently being used to treat short-stature disorders in patients to improve growth outcomes [21,22,23,24]. However, as described above (Section 2.4), exposing developing zebrafish embryos to exogenous CNP *inhibits* growth. To explain this discrepancy, we used a multiplex RT-qPCR assay to examine the effects of exogenous CNP on the expression of natriuretic peptide components in developing zebrafish embryos. The assay allowed simultaneous detection of transcripts encoding the four CNPs (*nppcl*, *nppc2*, *cnp3* and *nppc4*), other members of the natriuretic peptide family (*nppa* and *bnp*), the natriuretic peptide receptors (*npra*, *npr1a*, *nprb*, *nprc*), and of processing enzymes involved in the synthesis of mature natriuretic peptides (*corin*, *furin a* and *furin b*). Total RNA was extracted from zebrafish embryos following treatment with either 0 or 100 nM CNP for up to 72 hpf. As shown (Figure 5A), transcript expression for *nppcl* was significantly reduced in 100 nM CNP-treated zebrafish embryos compared to control embryos (to 0.43 ± 0.04 fold (*p* < 0.01) and 0.70 ± 0.09 fold (*p* < 0.05), at 48 hpf and 72 hpf, respectively). Whilst CNP treatment caused no significant changes in *nppc2* and *cnp3* transcripts, *nppc4* transcripts were modestly reduced within 24 hpf (to 0.72 ± 0.05 fold, *p* < 0.01), but markedly increased after 48 hpf exposure (to 1.75 ± 0.05 fold, *p* < 0.01). 

Exogenous CNP exposure caused a significant increase in transcripts for *nppa* (Figure 5E; 24 hpf, to 1.3 ± 0.06 fold, *p* < 0.05), increased *npra* at 24 hpf but inhibited by 48 hpf (Figure 5J; 24 hpf, to 1.5 ± 0.08 fold, *p* < 0.05; 48 hpf, to 0.6 ± 0.02 fold, *p* < 0.05), and decreased *npr1a* at 24 hpf but increased by 48 hpf (Figure 5K; 24 hpf, to 0.84 ± 0.02 fold, *p* < 0.01; 48 hpf; to 1.1 ± 0.03 fold, *p* < 0.05). In contrast, the expression of the processing enzymes, *corin* and *furina*, and the natriuretic peptide receptor *nprb*, were all significantly inhibited by CNP treatments within 24 hpf (*corin* 0.73 ± 0.5 fold, *p* < 0.001; *furina* 0.78 ± 0.02 fold, *p* < 0.01; *nprb* 0.94 ± 0.01 fold, *p* < 0.05). These data show exposure to exogenous CNP during zebrafish development can cause significant changes in natriuretic peptide gene expression, most notably by causing inhibition of *nppcl* expression. 

### 2.6. CRISPR/Cas9 Disruption of Nppcl Affects Growth in Developing Zebrafish Embryos

Having shown that exogenous CNP exposure results in significant disruption to the expression of multiple genes within the natriuretic peptide pathway, we next used the CRISPR/Cas9 system to transiently knock-down expression of endogenous *nppcl* in developing zebrafish embryos. Embryos were injected with either *nppcl* gRNA and Cas9 RNA, or Cas9 RNA alone; those displaying mild to moderate phenotypes (Figure 6A) were then reared to 24, 48 and 72 hpf in normal aquaria water, before euthanizing, and fixation in 4% (*w*/*v*) paraformaldehyde in preparation for phase-contrast microscopy and morphometric analysis. As shown (Figure 6B), CRISPR *nppcl*-treated embryos had significantly reduced head width at 48 and 72 hpf compared to control embryos (48 h: from 321.0 ± 10.51 µm to 285.0 ± 7.96 µm, *p* < 0.05, in untreated and CRISPR *nppcl*-treated embryos, respectively (*n* = 12); 72 h: from 411.9 ± 5.5 µm to 381.9 ± 6.5 µm, *p* < 0.01, in untreated and CRISPR *nppcl*-treated embryos, respectively (*n* = 12). Body length was significantly reduced at all time points in CRISPR *nppcl*-treated embryos compared to control embryos (24 h: from 1836 ± 46.18 µm to 1667 ± 42.94 µm, *p* < 0.05, in untreated and CRISPR *nppcl*-treated embryos, respectively (*n* = 12); and eye diameter was significantly reduced at 24 hpf and 72 hpf in CRISPR *nppcl*-treated embryos (24 h: from 154.6 ± 7.685 µm to 129.3 ± 6.556 µm, *p* < 0.05; 72 h: from 216.5 ± 3.376 µm, 199.0 ± 5.198 µm, *p* < 0.01, in untreated and CRISPR *nppcl*-treated embryos, respectively (*n* = 12). Collectively, these data suggest that reduced expression of *nppcl* during development results in growth impairments.

### 2.7. Effect of CRISPR/Cas9 Disruption of Nppcl on Expression of Natriuretic Peptide Components in Developing Zebrafish Embryos

Similar to our studies with exogenous CNP (Section 2.5, above), we next determined the effect of disrupting *nppcl* on the expression of natriuretic peptide gene expression in developing zebrafish embryos. Embryos were injected with either *nppcl* gRNA and Cas9 RNA, or Cas9 RNA alone; those displaying mild to moderate phenotypes were then reared to 24, 48 and 72 hpf in normal aquaria water, before euthanizing, and extracting total RNA in preparation for multiplex RT-qPCR. As expected, *nppcl* expression in CRISPR *nppcl*-treated embryos was significantly inhibited (to 0.37 ± 0.05 fold (*p* < 0.001), and 0.72 ± 0.06 fold (*p* < 0.05), compared to control embryos at 48 hpf and 72 hpf, respectively (*n* = 3; Figure 7A). In contrast, the expression of *nppc2* in CRISPR *nppcl*-treated embryos was markedly upregulated compared with control embryos (by 6.6 ± 0.4 fold (*p* < 0.001) and 3.4 ± 0.5 fold (*p* < 0.05) compared to control embryos at 24 hpf and 48 hpf, respectively (*n* = 3; Figure 7B). There was no significant change in the expression of *cnp3* in CRISPR *nppcl*-treated embryos (Figure 7C), but *nppc4* expression was moderately upregulated in CRISPR *nppcl*-treated embryos compared with control embryos (by 1.7 ± 0.23 fold (*p* < 0.05) and 1.23 ± 0.04 fold (*p* < 0.05) compared to control embryos at 24 hpf and 48 hpf, respectively (*n* = 3; Figure 7D). In terms of the expression of other natriuretic peptides, *nppa* was significantly reduced in CRISPR *nppcl*-treated embryos compared to control embryos (by 0.73 ± 0.01 fold (*p* < 0.05; *n* = 3; Figure 7E), whereas *bnp* expression was upregulated in CRISPR *nppcl*-treated embryos at all time points (to 2.02 ± 0.06 fold (*p* < 0.01), 1.4 ± 0.04 fold (*p* < 0.001), and 1.71 ± 0.05 fold (*p* < 0.01) compared to control embryos at 24 hpf, 48 hpf and 72 hpf, respectively (*n* = 3; Figure 7F–H). In contrast to the effects on natriuretic peptide transcripts, the expression of *corin*, *nprb*, *npra*, and *furinb* in CRISPR *nppcl*-treated embryos were unchanged compared with control embryos, with only modest changes to *furina* (24 hpf: to 0.83 ± 0.01 fold (*p* < 0.05; Figure 7H), *npr1a* (48 hpf: to 0.91 ± 0.02, *p* < 0.05; Figure 7K) and *nprc* (48 hpf: to 1.2 ± 0.02 fold (*p* < 0.01; Figure 7M) (*n* = 3). These data reveal that the expression of multiple natriuretic peptide components is altered in CRISPR *nppcl*-treated embryos.

## 3. Discussion

C-type natriuretic peptide has well-known growth-promoting effects in bone [10,12,36], and CNP analogues have recently been used as therapeutics in the treatment of short stature syndromes [21,22,23,24]. Multiple loss-of-function mutations have been reported in either the *NPPC* or *NPR2* genes that cause short-stature in humans [14,15,16], with *Nppc* and *Npr2* knock-out mouse models displaying dwarf phenotypes [10,11]. However, understanding of the role of CNP in development remains poor. Here, we have manipulated exposure to CNP during early development, through pharmacological and molecular approaches, using zebrafish as a model organism. Our data show that over-exposure to exogenous CNP during the first 72 hpf results in impaired growth, and developmental abnormalities such as reduced head width and eye diameter. Furthermore, the mechanism for this apparent paradox is likely to be the downregulation of endogenous *nppcl*, effectively resulting in a knock-down of CNP (through negative feedback), and subsequent growth inhibition. These data are the first to describe roles for CNP during embryonic development.

Our initial investigations aimed to characterize transcripts encoding CNP in the zebrafish genome (*nppcl*, *nppc2*, *cnp3*, *nppc4*), determine their homology and establish their spatial and temporal expression profile. Bioinformatic studies identified the presence of four CNP transcripts, and bootstrap analyses indicated that *nppcl* (CNP1/CNP-like) and *nppc4* formed a clade with other teleost CNP-1 sequences, and most likely represent zebrafish orthologues of mammalian CNP. In contrast, *nppc2* was more closely associated with other teleost *nppa* and *nppb* genes, and *cnp3* was closely associated with other teleost *cnp3* genes. Recent studies have identified additional *nppc4/cnp4* genes in teleost species (including from zebrafish), that show similar proximity to human CNP in their phylogenetic analyses [33]. The peptide sequence of zebrafish *nppcl* and human CNP share 87% homology, and within the disulfide ring structure that homology reaches 100%–this ring structure is a requirement for biological activity at the GC-B/Npr2 receptor [1,4]. In contrast, the disulfide ring peptide sequence of zebrafish *nppc4/cnp4* is 76% compared with human CNP. Our data reveal that disruption to zebrafish *nppcl* essentially phenocopies the growth disruption seen in mice and humans deficient in *NPPC/Nppc*/CNP (and mammals do not possess additional transcripts encoding CNP). This loss/reduction in *nppcl* leads to these growth disruptions, whilst expression of *nppc2, cnp3* and *nppc4* remains either unchanged or enhanced, suggesting that the product of the *nppcl* gene functionally performs a very similar role in zebrafish to that performed by *NPPC/Nppc* in humans and mice. Collectively, these observations provide evidence to support *nppcl* as being a functional orthologue of mammalian CNP, although further studies are needed to establish whether *nppcl* and *nppc4/cnp4* perform similar functions. 

Conventional end-point RT-PCR identified transcripts for all four CNP genes (*nppcl*, *nppc2*, *cnp3*, *nppc4*), *anp*, *nppb*, all three main natriuretic peptide receptors (*npra/npr1a*, *nprb*, *nprc*) and the processing enzymes (*corin*, *furin a*, *furin b*). Of the CNP transcripts, *nppcl* and *nppc4* were detected within 6 hpf in developing zebrafish embryos. At this stage in development, it is possible that this reflects some maternal transcript. Expression of the other CNP gene transcripts, as well as those encoding the other peptides, receptors, and processing enzymes, were detectable from 12 hpf. In situ hybridization for the four CNP transcripts revealed a spatial expression profile conserved predominantly within the central nervous system. A similar profile was observed for the CNP-specific *nprb* receptor, which was detected throughout the developing nervous system, with some expression with the GI tract (Appendix A). These expression patterns support a major role for CNPs in the development of the central nervous system, as has previously been suggested in rodents [5,37]. The multiple RT-qPCR assays enabled quantitative assessment of gene expression changes. Transcripts for *nppa* and *bnp* peaked at 24 hpf and remained constant thereafter; the expression profiles of the three natriuretic peptide receptors and natriuretic peptide processing enzymes were only modestly altered during the first 5 days of development. Of the CNP genes, transcripts for *nppc2* and *nppc4* displayed little variation in abundance between 24 hpf and 120 hpf, but *cnp3* expression increased approximately 5 fold at 96 hpf. In marked contrast, *nppcl* transcript expression increased rapidly to a peak of 46 fold at 120 hpf, compared to at 24 hpf. This dramatic increase in developmental expression of *nppcl* in zebrafish embryos strongly suggests that this CNP transcript is the predominant form in mediating the effects of CNP during embryogenesis, and further supports the bioinformatic prediction that *nppcl* is a functional orthologue of mammalian CNP (*Nppc*) in zebrafish. The developmental window we examined ranged from late segmentation phase through to larval phase, during which time all developmental landmarks will have been reached in zebrafish [38]. Without further investigation, it is difficult to determine the specific role of the natriuretic peptide system during zebrafish development. Nevertheless, our in situ hybridization data strongly support a role for the CNP peptides and the GC-B receptor in neurodevelopment. 

Having identified the spatial and temporal characteristics of various transcripts encoding the natriuretic peptide system in zebrafish, we next examined the potential role of CNP during development. The critical role of CNP in supporting appropriate growth has been well established, and CNP analogs are used as therapies for short stature disorders in humans [21,22,23,24]. However, very little is understood about the function of CNP throughout development. Using zebrafish as a vertebrate model, we examined the effects of CNP during early growth and development. Despite the growth-promoting effects of CNP in short stature patients [21,22,23,24], and the overgrowth phenotype seen in patients with activating mutations in either *NPR2* or gene duplication of *NPPC* [14,15,16], we observed an *inhibition* in growth and development in zebrafish exposed to CNP during early embryogenesis. Overall growth, head width, and eye diameter were all significantly inhibited in zebrafish embryos treated with CNP compared to those raised in normal aquaria water alone. These inhibitory effects of CNP on growth appear to be partially mediated by an increase in apoptosis, as TUNEL staining in zebrafish embryos treated with CNP was significantly increased by 72 hpf. CNP is known to be anti-apoptotic in conditions of renal injury [39] and in retinal ganglion cells [40], but loss of *Npr2* causes apoptosis in mouse Leydig cells [41]. Apoptosis is involved in the development of some growth-associated disorders, such as mucopolysaccharidoses (MPS) [42]. Interestingly, CNP is a potential therapeutic in treating MPS by improving endochondral ossification [43]. The cellular mechanism for the observed increase in apoptosis in CNP-treated zebrafish embryos is currently unknown, but it is more likely associated with a loss of CNP-mediated signaling, as our molecular characterization of these zebrafish embryos revealed a significant inhibition of *nppcl* expression. This downregulation in endogenous *nppcl* in response to exogenous CNP treatment is also accompanied by a downregulation in endogenous *nppc4* expression, which suggests that there is some functional linkage between these two CNP transcripts in zebrafish. It is possible that the subsequent increase in *nppc4* expression following exogenous CNP treatment is a compensatory mechanism to address the inhibition. Exogenous CNP treatment has recently been shown to cause downregulation in *Nppc* mRNA expression in cartilaginous tissue in rats, suggesting that systemic feedback might regulate the CNP system, at least in some tissues [44]. It remains to be established if our current observations in zebrafish are also due to direct feedback mechanisms. A further possibility is that prolonged exposure to exogenous CNP caused homologous desensitization of the GC-B receptor, as has been reported in multiple systems [45,46,47,48,49]. However, due to the apparent low efficacy of cGMP generation in our experimental system, the contribution of tachyphylaxis towards the downregulation phenotype needs further investigation. 

Exposure to exogenous CNP during early development in zebrafish revealed a disrupted growth phenotype that was potentially mediated via the downregulation of endogenous *nppcl*. To determine if this mechanism was responsible for the observed inhibition in growth, head width and eye diameter, we used the CRISPR-Cas9 system to target disruption of *nppcl*. The growth and developmental characteristics of these CRISPR *nppcl*-treated zebrafish essentially phenocopied that seen when treating zebrafish with exogenous CNP. As expected, there was a pronounced inhibition in endogenous *nppcl* expression (the targeted transcript), which was consistent with the downregulation seen following pharmacological intervention with CNP. However, in contrast to the molecular response to CNP treatment, zebrafish embryos treated with CRISPR *nppcl* showed a large upregulation of *nppc2* expression, accompanied by a modest increase in *nppc4* (summary data are presented in Appendix A). In addition, CRISPR *nppcl*-treated zebrafish embryos also demonstrated increased expression of *bnp*. Each of these responses could reflect a compensatory mechanism in the presence of inhibited *nppcl* expression. Whilst the growth and developmental phenotype of CNP-treated and CRISPR *nppcl-*treated zebrafish embryos are shared, the mechanisms underlying them are unclear. Certainly, the molecular response to both manipulations is not identical. The rapid, and significant upregulation of *nppc2* and *bnp* seen in CRISPR *nppcl*-treated zebrafish is not observed in CNP-treated embryos, and mouse models of *Nppc* deficiency have not reported compensation by other natriuretic peptides [10]. These differences in responsiveness could reflect the speed of action of pharmacological vs. genomic manipulation; although we could detect significant increases in cGMP production in primary cultures of zebrafish embryonic cells, the response was delayed and modest, suggesting relatively inefficient activation of the CNP-GC-B-cGMP pathway. In contrast, injection of conventional CRISPR constructs can have direct effects on the target genes within hours [50]. Our bioinformatic analyses showed a closer evolutionary relationship between *nppcl* and *nppc4*, but functional relationships between *nppcl* and *nppc2* cannot be ruled out, on the basis of our current findings. The functional role of the other CNP transcripts (*nppc2*, *cnp3* and *nppc4*) requires further investigation. Paralogues of *nppc4* have been cloned in both zebrafish and eel (named *cnp4a* and *cnp4b*), that are expressed in the brain, pituitary, and peripheral tissues such as spleen and intestines [33]. Targeted disruption of these additional CNP paralogues will provide a clearer understanding of their functional role in fish.

The original mouse models of disrupted *Nppc* or *Npr2* expression described dwarfism and early death phenotypes [10,11]. However, these mice did not display the impaired growth characteristics at birth but developed them post-partum. In contrast, our data show that disrupted endogenous *nppcl* expression during development in zebrafish leads to stunted growth characteristics. A major difference between the mouse and zebrafish models is the influence of maternal CNP on the development of *Nppc* knock-out mice in utero; as zebrafish embryos do not receive a maternal supply of CNP during development, it is likely that observed gene expression changes (such as altered expression of other natriuretic peptides and receptors) reflects an attempt to compensate for reduced levels of endogenous *nppcl*. No such compensation was described in mouse models with reduced CNP expression [10].

Although our data are the first to demonstrate a role for *nppcl* in zebrafish development, there are several limitations to the current studies. Since we initiated our investigations, subsequent paralogues of *nppc4* have been identified, which through phylogenetic analyses show a strong resemblance to mammalian CNP [33]. However, comprehensive reverse genetics approaches have not yet been performed that would help distinguish whether the different CNP transcripts in zebrafish perform different functions. A further limitation of the current study is our use of mammalian CNP22 as an exogenous source of CNP. Zebrafish-specific *nppcl* peptide may exhibit different biological activity compared with mammalian CNP, especially in the presence of native GC-B/Npr2 receptors. Despite this limitation, the disulfide ring structure of the peptides encoding zebrafish *nppcl* and human *NPPC* are identical, and these ring structures are essential for biological activity [1,4]. Future pharmacological investigations of zebrafish CNP acting on zebrafish GC-B receptors would improve our understanding of natriuretic peptide signaling in fish species. Finally, we have only investigated the consequences of disrupting *nppcl* during the first 96 hpf; it is important to establish whether the observed growth deficiencies are maintained into adulthood, or whether the increased expression of other CNP transcripts are able to compensate.

There are several CNP analogues that are being used to treat patients with short stature conditions (typically achondroplasia), including Vosoritide and TransCon CNP [21,22,23,24]. Although our current data suggest that treatment with exogenous CNP can result in disrupted growth, it is important to emphasize that we have focused on developmental exposure to exogenous CNP. In contrast, the current CNP therapies are for use in children from 5 yrs of age [23]. Consequently, our findings do not provide any evidence that contradicts the successful use of exogenous CNP therapies to treat short stature patients. 

In conclusion, we demonstrate an early developmental role for CNP in zebrafish that determines appropriate growth characteristics that had not been revealed by previous genetic disruption studies in mice [10,44,51,52]. Furthermore, zebrafish embryos are sensitive to exogenous CNP stimulation, which leads to downregulation of endogenous *nppcl* expression, resulting in similar growth abnormalities that are apparent in CRISPR *nppcl*-treated embryos. These data provide further evidence of an important role for CNP in regulating growth in vertebrate species, and indicates that intrinsic control of CNP is important for appropriate developmental growth. 

## 4. Materials and Methods

### 4.1. Bioinformatics, Phylogenetic Sequence Analyses and CRISPR Design

Genetic sequences of selected zebrafish genes, both known protein coding and predicted, were discovered and obtained from the zebrafish genome assembly-v9 by searching the Ensembl genome browser (http://www.ensembl.org/Danio_rerio/Info/Index, accessed on 1 January 2015) and NCBI Gene (http://www.ncbi.nlm.nih.gov/gene, accessed on 1 January 2015). These reference databases were also used to obtain genetic sequences for the other species used in this project. Whole genetic sequences were compared using multiple sequence alignments performed by the EMBL-Ebi alignment tool ClustalW2 (http://www.ebi.ac.uk/Tools/msa/clustalw2/, accessed on 1 January 2015) and phylogenetic trees inferred using the molecular evolutionary genetic analysis programme MEGA5 (http://megasoftware.net). Phylogenetic trees were inferred using the maximum-likelihood method adopting the Tamura–Nei substitution model and supported by bootstrap analysis of 1000 replicates. Zebrafish protein sequences were obtained from obtained from the NCBI Protein database (http://www.ncbi.nlm.nih.gov/protein). Amino acid sequences were aligned and sequence homology calculated using ClustalW2 (http://www.ebi.ac.uk/Tools/msa/clustalw2/).

The whole genomic coding sequence for zebrafish *nppcl* (ENSDARG00000043460) was inserted into the CRISPR design software E-CRISP (http://www.e-crisp.org/E-CRISP/, accessed on 1 January 2015) and the palindromic repeat search parameters adjusted to their most stringent settings. The program provided an output list of all suitable target sites across the genetic sequence and scored them on their specificity, annotation, efficacy and proximity to the 5′ end. The process was repeated with a comparable CRISPR design program, the MIT Optimized CRISPR design tool (http://crispr.mit.edu, accessed on 1 January 2015) using similar design parameters. The top scoring design sequences across from both design lists with zero off-target interactions were then evaluated and the top scoring design present in both lists was selected. Oligonucleotide primers were subsequently designed using this target sequence with palindromic repeat sequences at each end and cloned into the pT7-gRNA plasmid as previously described [53]. Successful cloning of these oligonucleotides into the pT7-gRNA plasmid would then act as the template for which specific gRNAs were transcribed. Appendix A shows the design and sequence of the *nppcl* gRNA.

### 4.2. Zebrafish Husbandry and Treatments

Tupfel-longfin (TL) zebrafish were maintained in an open-cycle aquarium at 28 °C on a 14:10 h light:dark cycle to mimic their natural environment. Embryos were captured from staged-adult pairings and reared in the dark at 28 °C in zebrafish aquaria water, which was replaced every 24 h. When embryos were to be captured at the desired developmental time point, developmental staging was confirmed [38]. All procedures were carried out in accordance with the UK Home Office Animals Scientific Procedures Act (1986). 

For exogenous CNP experiments, embryos were collected from adult pairings immediately post-fertilization and divided into two separate sample groups with approximately 50 embryos in each. A 100 nM solution of CNP (Sigma-Aldrich, Poole, UK) was made up in zebrafish aquaria water. Zebrafish aquaria water was then removed from both sample groups of embryos and replaced with either the 100 nM CNP solution for the treatment group or fresh zebrafish aquaria water in the control group. Embryos were reared under normal conditions and aquaria water with or without CNP was and replaced with fresh every 24 h. Embryos were later captured at 24, 48, 72, 96 or 120 hpf depending on desired developmental stage. 

Zebrafish embryos were collected at 24, 48 or 72 hpf, dechorionated and euthanized using zebrafish euthanasia solution. The embryos were then washed for 3 × 5 min with PBST and fixed in 4% (*w*/*v*) PFA overnight at 4 °C. Post-fixation the embryos were washed for 2 × 5 min in PBS and mounted onto glass microscope slides in a 20% (*v*/*v*) solution of glycerol in PBS. Multiple images were then taken along or across the embryos at 20 × magnification in either an anteroposterior or dorsal plane using phase microscopy using an Axiovert 135 inverted microscope (Carl Zeiss Ltd., Cambridge, UK) and DC500 color camera. Corresponding images from each embryo were then imported into Adobe Photoshop (Adobe, USA) and Z-stacked to form a single image. Biometric measurements of embryo length, head width and eye diameter were subsequently taken using Image J (http://imagej.nih.gov/in/).

### 4.3. RNA Extraction, RT-PCR and Wholemount In Situ Hybridization

Pools of total RNA were extracted from 50 zebrafish larvae using RNAbee (AMS Biotechnology, Abingdon, UK) as previously described [49], and subjected to DNase treatment using Qiagen RNAeasy kit, in accordance with manufacturer’s instructions. RNA concentrations were determined using ND-100 spectrophotometer (Nanodrop^TM^), and 260/280 ratios were used to maintain RNA integrity. First-strand cDNA synthesis was performed using a High Capacity cDNA reverse transcriptase kit (Thermo Fisher, Horsham, UK), before conducting PCR for natriuretic peptide system genes, using the specific primers listed in Appendix A. 

Some PCR products were cloned and used to construct digoxigenin labelled RNA probes, using a Digoxigenin RNA labelling mix (Roche, Welwyn Garden City, UK). in situ hybridization was performed with probes specific for each of the CNP orthologues (*nppcl*, *nppc2*, *cnp3*, *nppc4*) as well as for *npr2*, using protocols described previously [54]. 

### 4.4. GeXP Multiplex RT-qPCR

Customized GeXP multiplex assays were designed to genes that included the natriuretic peptide system (see Appendix A). Target-specific reverse transcription (using 100 ng RNA as template), and PCR amplification were performed as we described previously [49,55,56], and in accordance with manufacturer’s instructions (Beckman Coulter, High Wycombe, UK). In brief, a master mix was prepared for reverse transcription as detailed in the GeXP starter kit (Beckman Coulter, High Wycombe, UK) and performed using a G-storm GS1 thermal cycler (GStorm Ltd., Somerset, UK), using the following protocol: 48 °C 1 min; 42 °C 60 min; and 95 °C 5 min. From this an aliquot of each reverse transcriptase reaction was added to PCR master mix, consisting of GenomeLab kit PCR reaction mix and Thermoscientific Thermo-Start Taq DNA polymerase. PCR reactions were performed using G-Storm GS1 thermal cycler with a 95 °C activation step for 10 min, followed by 35 cycles of 94 °C 30 s; 55 °C 30 s; 70 °C 60 s. Products were separated and quantified using CEQTM 8000 Genetic Analysis System, and GenomeLab Fragment Analysis software (eXpress Analysis Version 1.0.25, Beckman Coulter, High Wycombe, UK).

### 4.5. Locomotion Assays

Locomotion assays were performed with individual TL/Wt zebrafish larvae that had been incubated in aquaria water containing either 0 or 100 nM CNP for XXX, with movement characteristics determined as described previously [57], using Ethovision XT (Noldus) software ver.7.0 (Tracksys Ltd., Nottingham, UK).

### 4.6. Wholemount TUNEL Staining

Zebrafish embryos were collected at 24, 48 or 72 hpf, their chorions removed and euthanized using zebrafish euthanasia solution. Embryos were transferred to 5 mL bijous and fixed in 4% (*w*/*v*) PFA overnight at 4 °C. PFA solution was removed and the embryos washed 3 times for 5 min with PBS. Subsequent to PBS washes, embryos were dehydrated using a MeOH gradient wash; 50% (*v*/*v*) MeOH for 5minutes, 100% (*v*/*v*) MeOH for 5 min, replace 100% (*v*/*v*) MeOH and place larvae at −20 °C overnight. Zebrafish embryos were rehydrated using a 50% (*v*/*v*) MeOH:50% (*v*/*v*) PBST wash for 5 min, followed by 3 times for 5 min with PBST. Embryo pigmentation was then bleached using zebrafish bleaching solution and incubation under a strong white light until no pigment was visible. Embryos were then permeabilized using Proteinase K at either: 1 ug/mL in PBST for 30 min (24 hpf), 1 ug/mL in PBST for 60 min (48 hpf) or 1.5 ug/mL in PBST for 60 min (72 hpf). Immediately after, the Proteinase K is removed and the embryos rinsed twice and washed once for 5 min with PBST. Before re-fixing with 4% (*w*/*v*) PFA for 20 min followed by a further PBST washes. Following this, embryos were immersed in pre-chilled 100% (*v*/*v*) EtOH:Acetone at a ratio of 2:1, respectively and incubated at −20 °C for 10 min, before being washed a further 3 times for 5 min with PBST. After removing as much PBST as possible, zebrafish were incubated in equilibration buffer for 1 h at RT. In the meantime, terminal deoxynucleotidyl transferase (TdT) reaction buffer was made by adding 2/3 reaction buffer 2 to 1/3 TdT and 0.3% Tritonx100. The majority of equilibration buffer was removed post-incubation, leaving the embryos submerged in as little as possible. 20ul reaction buffer was then added and the embryos incubated overnight at 37 °C. The following day, reaction buffer was removed and the embryos washed in PBST for 6 × 10 min at 37 °C, followed by 6 × 10 min washes at RT. Embryos were then blocked in MAB Block for 3 h at RT before incubation with anti-Digoxigenin-AP antibody, made up in fresh MAB Block at 1:3000, overnight at 4 °C. 

Following this overnight incubation, the embryos were further washed with PBST for 4 × 20 min incubations at RT. AS much PBST was then removed and the embryos allowed to equilibrate in NTMT buffer for 20 min. NTMT was then replaced by NTMT-NBT/BCIP and left at RT until the reaction had developed. Once staining was clearly visible, the NTMT-NBT/BCIP was removed, the embryos washed in PBST and fixed in 4% (*w*/*v*) PFA overnight at 4 °C. Post-fixation the embryos were washed for 2 × 5 min in PBST and stored in a 20% (*v*/*v*) solution of glycerol in PBS until mounted on glass slides. Images were captured at 5× magnification using bright-field microscopy and the relative quantities of TUNEL staining analyzed using a Volocity 3D image analysis software (Ver. 6.3) (PerkinElmer, Waltham, MA, USA) and customized measurement parameters. The mean number of positively stained pixels were calculated and statistical comparisons performed.

### 4.7. CRISPR/Cas9 Disruption of Zebrafish Embryos

In vitro transcribed *nppcl/nppcl*-pT7-gRNA and pCS2 Cas9 RNA (Eurofins, Ebersberg, Germany) were thawed and diluted to give working concentrations of co-injectable RNA with 10× CRISPR buffer (final concentration of 1×) and Type 1+ ultrapure deionized water. Working concentrations of RNA were aliquoted and stored at −80 °C until required to prevent freeze–thaw degradation of the RNA. When required, working aliquots of RNA were thawed on ice and mixed thoroughly. Glass microinjection needles were produced concurrently to reduce the likelihood of contaminating the needles. Glass needles were backloaded using an Eppendorf Microloader pipette tip (Hamburg, Germany) with 1.5 µL of CRISPR/Cas9 RNA solution. Needles were inserted into a tri-plane micromanipulator, the needles clipped and the RNA droplet size calibrated. The microinjection set up was then adjusted to deliver approximately 1nl of co-injectable RNA per injection. Embryos were collected and processed in the same manner, immediately after fertilization and approximately 50 were aligned against the edge of a microscope slide placed onto a Petri dish lid. CRISPR/Cas9 RNA was subsequently injected directly into the cytosol of 1-cell stage embryos alongside controls of *nppcl/nppcl*-pT7-gRNA or pCS2 Cas9 RNA only. Injected embryos were afterwards reared under normal conditions up to 72 hpf and their phenotypes monitored closely. Initial experiments were performed to optimize titres of gDNA and Cas9, by screening injected embryos for phenotypes. Severe phenotypic changes included reduced embryo length, often with an abnormally curled tail, impaired cranial and brain formation, severe micropthalmia (reduced eye width) and pericardial oedema. The optimal concentrations for generating CRISPR *nppcl* mutants was 175 ng/µL gRNA:150 ng/µL Cas9 RNA, as this yielded the highest proportion of viable embryos with either mild or moderate phenotypes. Any embryos displaying severe phenotypes were removed from the pool of injected embryos at 24 hpf and euthanized. 

### 4.8. Embryonic Zebrafish Primary Culture and cGMP Enzymeimmunoassay

TL/Wt zebrafish embryos were reared under normal conditions to 72 hpf at which point they were collected and dechorionated if necessary. Embryos were transferred to zebrafish anesthesia solution and incubated on ice for 30 min. Anesthesia solution was removed by rinsing briefly in Calcium chloride-free Ringers solution. Ringers solution was removed and embryos incubated in custom ATV solution containing 10% (*v*/*v*) Trypsin at 28 °C in a sterile environment. Embryos were regularly triturated with a narrow bore pipette to aid dispersal and the dissociation monitored regularly under the microscope. Once majority of dissociation was single cells, 100 mM calcium chloride and fetal calf serum was added to stop the enzymatic reaction. The dissociation mixture was centrifuged at 300× *g* for 5 min, the supernatant discarded, and pelleted cells resuspended in 15 mL L-15 supplemented medium. Centrifugation was repeated and supernatant discarded. Cells were resuspended in L-15 growth medium plated and plated at 250,000 cells per well in a 24-well cell culture plate. Plate was centrifuged at 300× *g* for 3 min and placed in a sterile incubator at 28 °C for 24 h. 

After the 24 h incubation, a 3 h reverse CNP time course was set up. Cells were treated with 100 mM CNP with 0.1 mM IBMX in physiological saline solution (PSS) for either 3 h, 1 h or 30 min. L-15 growth medium was removed and 200 µL of CNP/IBMX in PSS applied at designated time points and cells returned to the 28 °C incubator. Upon completion the 700 µL 100% (*v*/*v*) EtOH pre-chilled at −20 °C was added to each well and the contents stored at −20 °C. Samples were transferred to 1.5 mL reaction tubes and placed in a freeze-dryer to lyophilize at 30 °C for 2 h or until dry. The samples were then analyzed for cGMP accumulation, using a commercially available enzymeimmunoassay (R&D Systems, Abingdon, UK) as described previously [48,49]. 

### 4.9. Data Presentation and Statistical Analyses

Multiplex RT-qPCR data are shown as the means ± SEM, typically from 6 to 8 independent RNA extractions (unless stated otherwise). Each RNA extraction was prepared from a pool of 50 zebrafish embryos (see Section 4.3). Numerical data were subjected to two-tailed unpaired *t*-tests, using in-built equations in GraphPad Prism 10.0 for Mac (GraphPad, San Diego, CA, USA). Immunoassay data were also analyzed using in-built sigmoidal nonlinear curve-fitting equations in GraphPad Prism 10.0 for Mac. Morphometric data are shown as box-and-whisker plots of medians (with min-max values), analyzed by a paired *t*-test (2-tailed), where *n* represents individual zebrafish embryos. Qualitative data from in situ hybridization are representative of images collected from 12 to 15 individual zebrafish embryos.

## Figures and Tables

**Figure 1 ijms-24-12921-f001:**
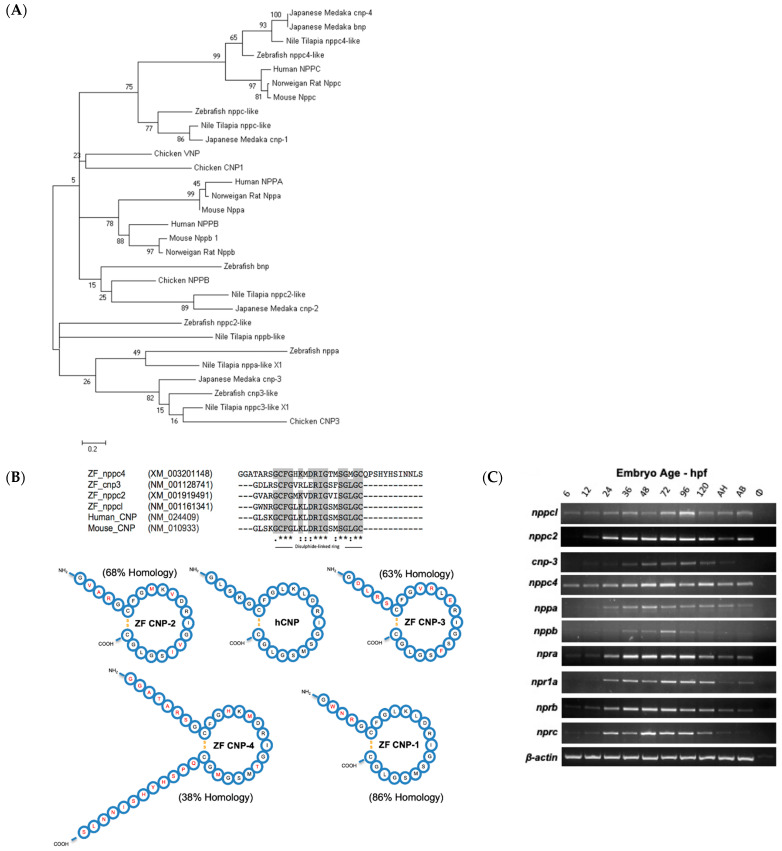
(**A**–**C**) Structural and molecular analyses of zebrafish C-type natriuretic peptides. (**A**) Multiple sequence alignments of published whole-natriuretic-peptide precursor sequences were aligned using Clustal Omega (http://www.ebi.ac.uk/Tools/msa/clustalo). Evolutionary genetic analysis was performed on the resulting alignment using MEGA5 (http://megasoftware.net) and a phylogenetic tree created using the maximum-likelihood method and Tamura–Nei substitution model. Nonparametric bootstrap sampling was performed (1000 replicates) alongside and shown as a percentage at each corresponding node. Scale bar represents a genetic distance of 0.2. (**B**) Zebrafish C-type natriuretic peptide, CNP1 (nppc-like), CNP2 (nppc2-like), CNP 3, (cnp3-like) and CNP 4 (nppc4-like), amino acid sequences were obtained from NCBI Protein (http://www.ncbi.nlm.nih.gov/protein) and processing endopeptidase cleavage sites identified. Predicted mature zebrafish CNPs were aligned with human and murine CNP using Clustal Omega (http://www.ebi.ac.uk/Tools/msa/clustalo) to assess structural similarities. Resulting sequence alignment with regions of high or absolute conservation highlighted in grey. Structural representation of known and predicted mature peptides. Amino acids that differ from human CNP are highlighted in red and calculated sequence homology with human CNP displayed as a percentage next to each zebrafish peptide. (**C**) Tupfel-long fin (TL) wild-type zebrafish embryos were captured at 6, 12, 24, 36, 48, 72, 96 and 120 h post-fertilization (hpf) along with adult zebrafish head (AH) and torso (AT) segments and harvested for total RNA extraction. cDNA was subsequently generated at 100 ng/µL for each developmental time point and used in PCR reactions with oligonucleotide primers designed against zebrafish *nppc*-like (CNP1) (product size: 602 bp), *nppc2*-like (CNP2) (332 bp), *cnp3*-like (CNP3) (557 bp), *nppc4*-like (CNP4) (379 bp) *npra* (GC-A) (352 bp), *npr1a* (GC-A) (640 bp), *nprb* (GC-B) (429 bp), *nprc* (NPR3) (748 bp), *nppa* (ANP) (602 bp), *nppb* (BNP) (332 bp) and *β-actin* (410 bp) alongside a no template control (ø). PCR products were electrophoresed on a 1.6% (*w*/*v*) agarose gel and visualized using ethidium bromide on a Bio-Rad ChemiDoc MP system (Bio-Rad, Hercules, CA, USA).

**Figure 2 ijms-24-12921-f002:**
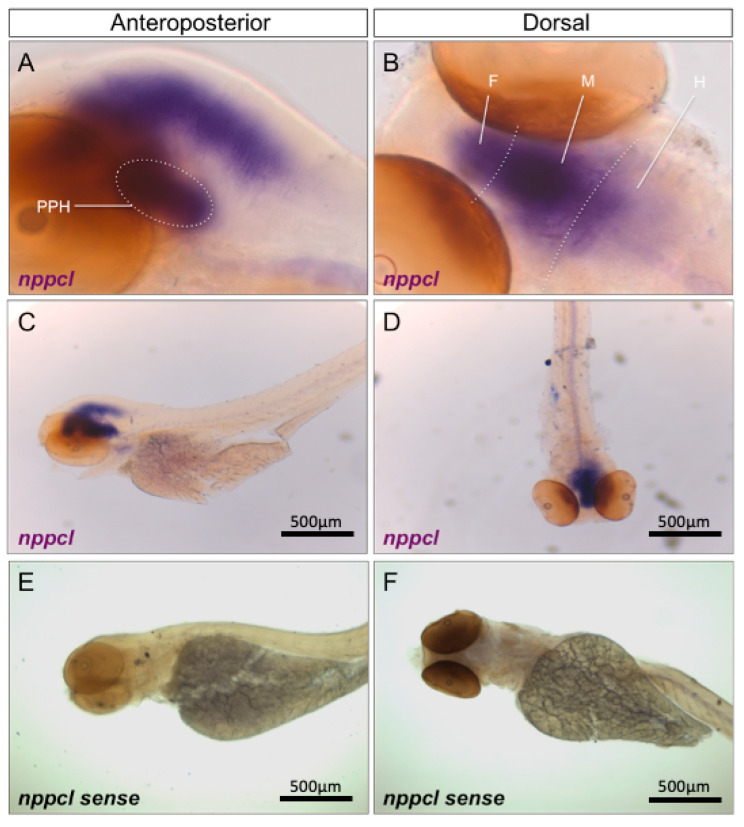
(**A**–**F**) Wholemount in situ hybridization of *nppcl* in 96 hpf wild-type zebrafish embryos. Zebrafish embryos were captured at 96 hpf and fixed with 4% PFA before wholemount in situ hybridization was performed to assess spatial expression of *nppcl* alongside sense probe controls. Images were captured at 20× (**A**,**B**) and 5x magnification (**C**–**F**) using bright-field microscopy on an Axiovert 135 inverted microscope (Carl Zeiss Ltd., Cambridge, UK) and DC500 color camera. In situ hybridization was performed using 12–15 embryos for each RNA probe and images are representative of the staining observed in multiple embryos. (PPH—presumptive pituitary and hypothalamus; F—forebrain; M—midbrain; H—hindbrain).

**Figure 3 ijms-24-12921-f003:**
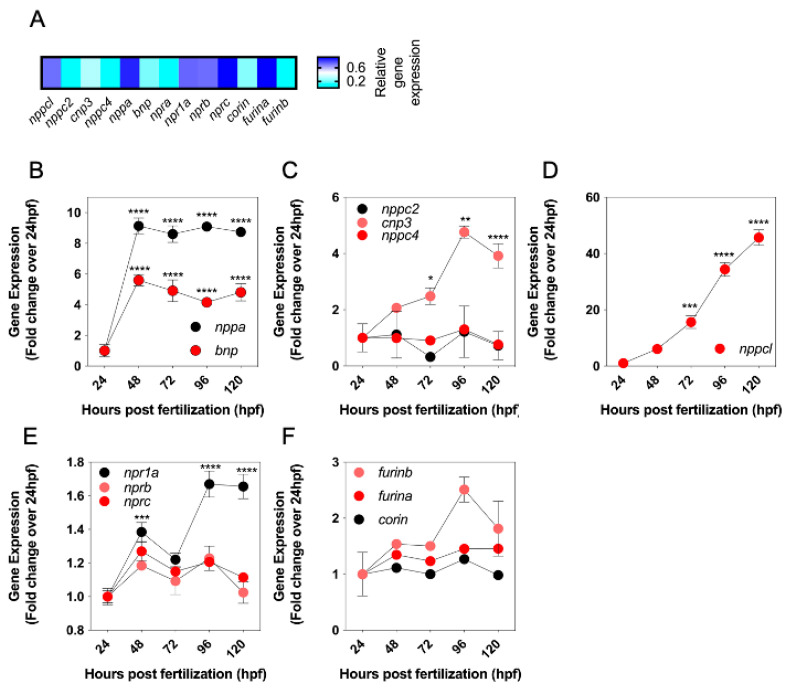
(**A**–**F**) Expression levels of natriuretic peptide system genes in developing wild-type zebrafish embryos. (**A**) Total RNA was extracted from whole TL/Wt zebrafish embryos captured at 72 hpf and normalized to 100 ng/µL. Target-specific cDNA synthesis performed followed by multiplex RT-qPCR examining the simultaneous expression of components of the zebrafish natriuretic peptide system (*nppc4*, *nppcl*, *nppc2*, *cnp3*, *nppa*, *bnp*, *npra*, *npr1a*, *nprb*, *nprc*, *corin*, *furina* and *furinb*). Products were separated using capillary gel electrophoresis and expression quantified using the GenomeLab GeXP analysis system. (**B**–**F**) Temporal expression of zebrafish natriuretic peptide pathway genes during development. Total RNA was extracted from whole TL/Wt zebrafish embryos captured at 24, 48, 72, 96 or 120 hpf and normalized to 100 ng/µL. Target-specific cDNA synthesis performed followed by multiplex RT-qPCR examining the simultaneous expression of components of the zebrafish natriuretic peptide system (**B**: *nppa*, *bnp*; **C**: *nppc2*, *cnp3*, *nppc4*; **D**: *nppcl*; **E**: *npr1a*, *nprb*, *nprc*; **F**: *corin*, *furina*, *furinb*). Products were separated using capillary gel electrophoresis and expression quantified using the GenomeLab GeXP analysis system. Data shown are normalized to the housekeeping gene (elfα) and presented as the mean ± SEM from 6 to 8 independent RNA extractions (except for *furin b*, where *n* = 4); * *p* < 0.05, ** *p* < 0.01, *** *p* < 0.001, **** *p* < 0.0001, significantly different from 24 hpf.

**Figure 4 ijms-24-12921-f004:**
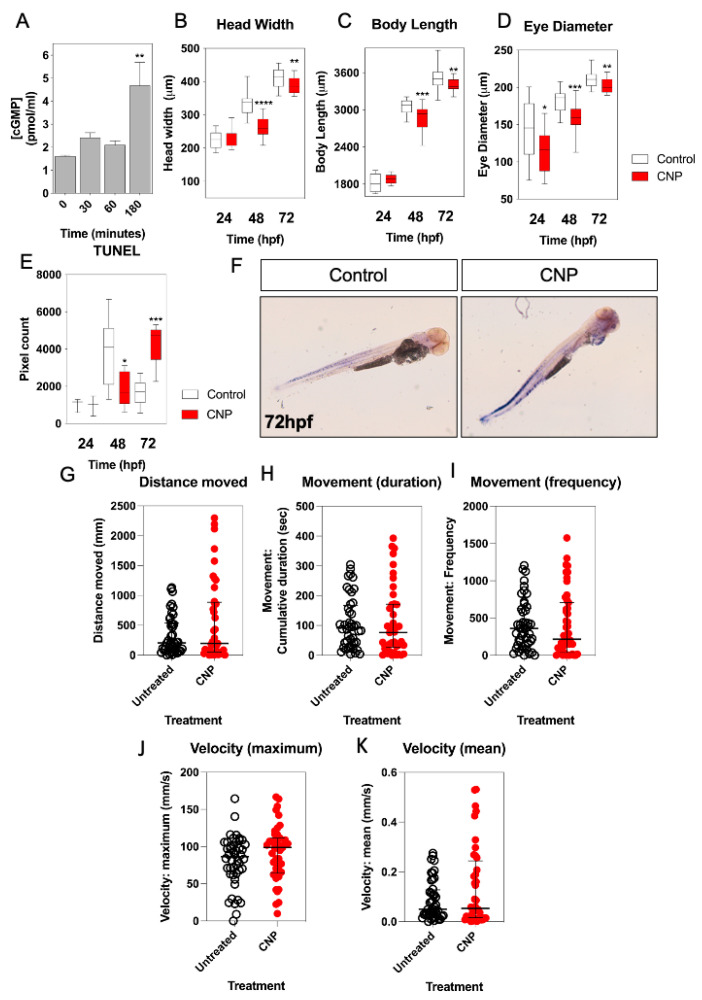
(**A**–**K**) Morphometric and locomotion analyses of TL/Wt zebrafish embryos exposed to exogenous CNP. (**A**) Total cGMP accumulation was measured in primary cultures of dispersed zebrafish embryos. Briefly, 250,000 cells/well were plated in 24-well plates and left to adhere for 24 h. Dispersed primary cell cultures were stimulated with physiological saline solution (PSS) containing 0 or 100 nM CNP in the presence of 1 mM isobutylmethylxanthine (IBMX) for up to 180 min. Stimulations were terminated with 100% (*v*/*v*) ice-cold EtOH, and samples extracted before assaying for total cGMP. Data shown are the means ± SEM of 3 independent stimulations (** *p* < 0.01, significantly different to control (0 nM CNP)). (**B**–**D**) TL/Wt zebrafish embryos, which had been reared in either 0 (control) or 100 nM CNP, were captured between 24 hpf, 48 hpf and 72 hpf, dechorionated and fixed with 4% (*w*/*v*) PFA. Embryos were subsequently mounted and 20× magnification phase-contrast images captured using an Axiovert 135 inverted microscope (Carl Zeiss Ltd., Cambridge, UK) and DC500 color camera. Phase-contrast images were Z-stacked using Adobe Photoshop (Adobe, San Jose, CA, USA) and morphometric measurements collected using ImageJ (http://imagej.nih.gov/in/, accessed on 1 January 2015). Measurements were taken for: embryo length (*n* = 12), head width (*n* = 12) and eye diameter (*n* = 12). Data shown are box-and-whisker plots of medians (with min-max values), analyzed by a paired *t*-test (2-tailed); * *p* < 0.05, ** *p* < 0.01, *** *p* < 0.001, **** *p* < 0.0001, significantly different from control. (**E**) TUNEL assays were performed on control and CNP-treated zebrafish embryos at 24 hpf (*n* = 3), 48 hpf (*n* = 8) and 72 hpf (*n* = 5); * *p* < 0.05, *** *p* < 0.001, significantly different from control. (**F**) Representative live images for untreated and CNP-treated embryos were captured using and a Nikon 159 dsZ2mv color camera, objective magnification was 10× and variable stage magnification at 3×. (**G**–**K**) Locomotion assays were performed over 24 h, with 72 hpf TL/Wt zebrafish larvae, which had been reared in either normal aquaria water or exposed to 100 nM CNP, to determine distance moved, movement duration, movement frequency, mean and maximum velocity. Data shown are dot plots, with medians and interquartile ranges calculated from control or CNP-treated larvae (*n* = 48).

**Figure 5 ijms-24-12921-f005:**
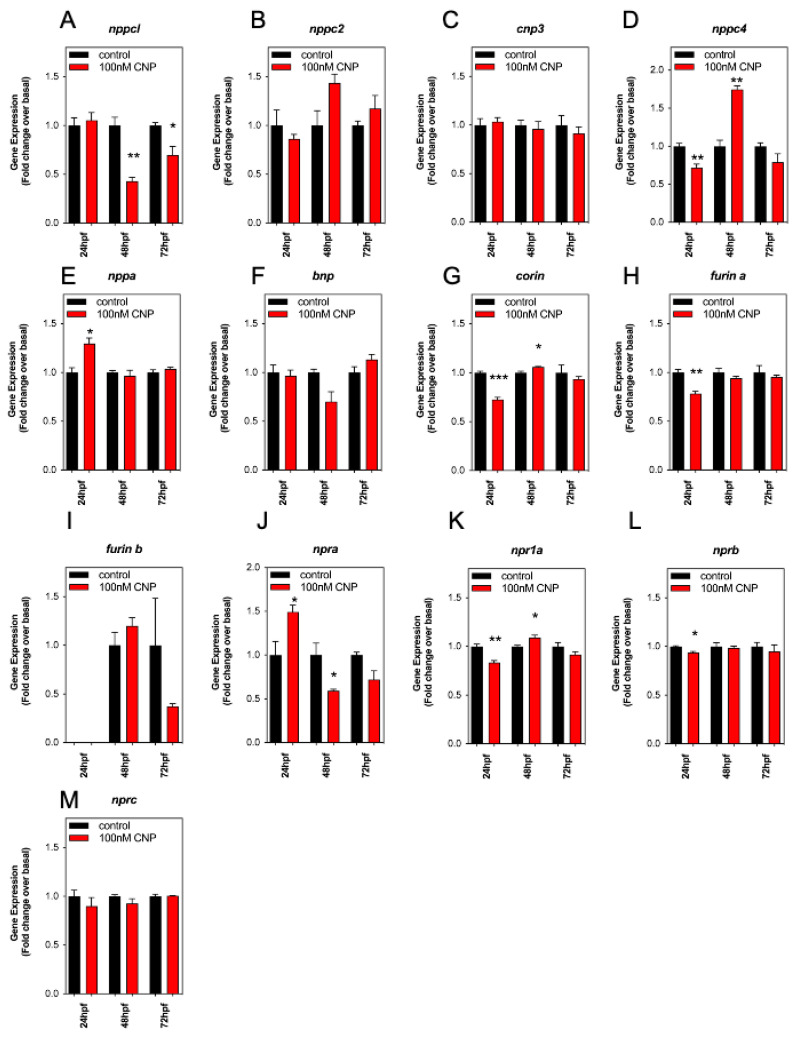
(**A**–**M**) RT-qPCR analysis of C-type natriuretic peptide system genes during development in Wt zebrafish embryos treated with exogenous CNP. Total RNA was extracted from whole TL/Wt zebrafish embryos, which had been reared in either normal aquaria water or exposed to 100 nM CNP, captured at 24, 48, or 72 hpf (*n* = 3 for each time point) and normalized to 100 ng/µL. (**A**–**M**) Target-specific cDNA synthesis was performed followed by multiplex RT-qPCR examining the simultaneous expression of components of the zebrafish natriuretic peptide system (*nppc4*, *nppcl*, *nppc2*, *cnp3*, *nppa*, *bnp*, *npra*, *npr1a*, *nprb*, *nprc*, *corin*, *furina* and *furinb*). Products were separated using capillary gel electrophoresis and expression quantified using the GenomeLab GeXP analysis system. Data were normalized to the housekeeping gene (*elfα*) and the mean displayed as fold change over basal expression found in untreated embryos, * *p* < 0.05, ** *p* < 0.01, *** *p* < 0.001, significantly different from control-treated embryos (*n* = 3 independent experiments).

**Figure 6 ijms-24-12921-f006:**
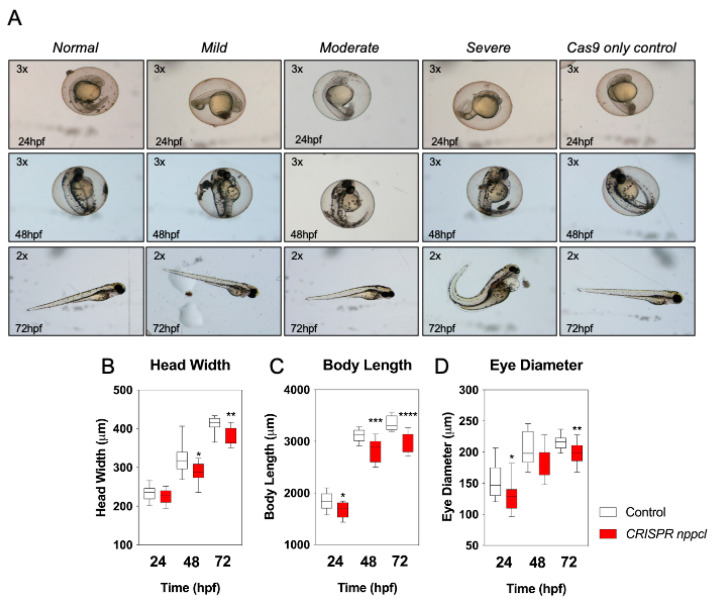
(**A**–**D**) Morphometric analysis of TL/Wt zebrafish embryos injected with *nppcl*/gRNA/Cas9 RNA. Zebrafish mutants co-injected with *nppcl*-gRNA and Cas9 RNA were reared alongside embryos injected with Cas9 RNA only and collected at 24 hpf, 48 hpf and 72 hpf, dechorionated and fixed with 4% (*w*/*v*) PFA. Embryos were subsequently mounted and 20x magnification phase-contrast images captured using an Axiovert 135 inverted microscope (Carl Zeiss Ltd., Cambridge, UK) and DC500 color camera. Phase-contrast images were Z-stacked using Adobe Photoshop (Adobe, USA) and morphometric measurements collected using ImageJ (http://imagej.nih.gov/in/). (**A**–**C**) Measurements were taken for: embryo length (*n* = 12), head width (*n* = 12) and eye diameter (*n* = 12). (**D**) Representative live images for were captured using a Nikon SMZ1500 stereomicroscope and a Nikon ds-2mv color camera, objective magnification was 10× and variable stage magnification at 3×. Data shown are box-and-whisker plots of medians (with min-max values), analyzed by a paired *t*-test (2-tailed); * *p* < 0.05, ** *p* < 0.01, *** *p* < 0.001, **** *p* < 0.0001, significantly different from control.

**Figure 7 ijms-24-12921-f007:**
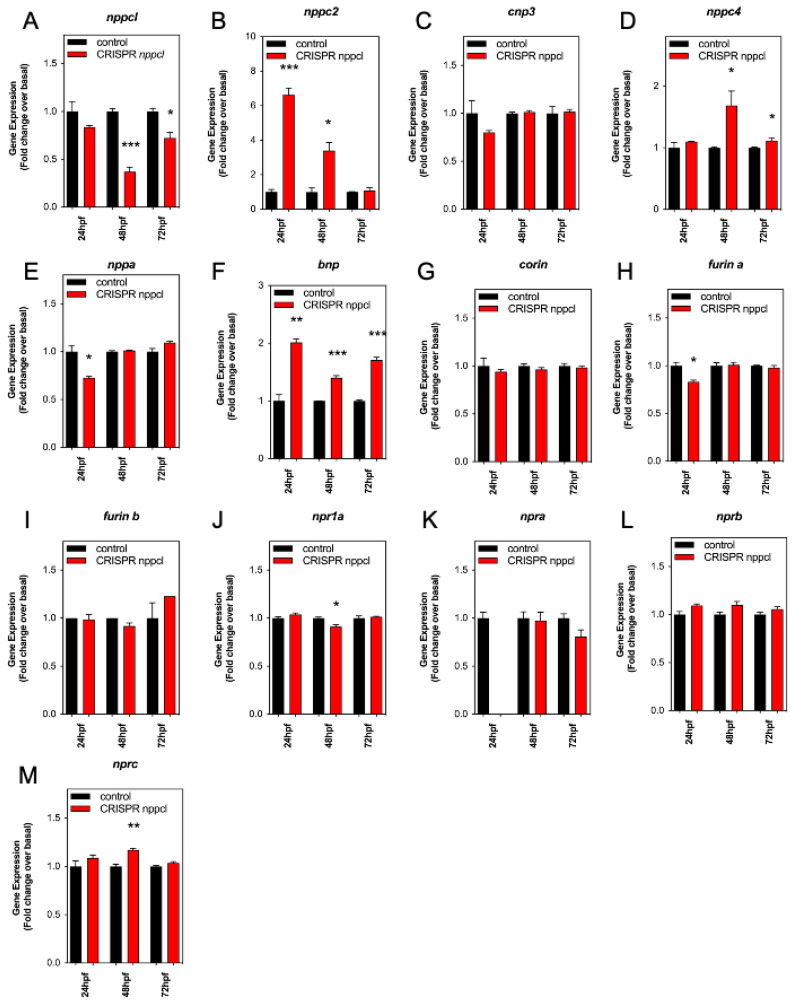
(**A**–**M**) RT-qPCR analysis of C-type natriuretic peptide system genes during development in Wt zebrafish embryos embryos injected with *nppcl*/gRNA/Cas9 RNA. Total RNA was extracted from whole TL/Wt zebrafish embryos, which had been reared in either normal aquaria water or exposed to 100 nM CNP, captured at 24, 48, or 72 hpf (*n* = 3 for each time point) and normalized to 100 ng/µL. (AZD) Target-specific cDNA synthesis was performed followed by multiplex RT-qPCR examining the simultaneous expression of components of the zebrafish natriuretic peptide system (*nppc4*, *nppcl*, *nppc2*, *cnp3*, *nppa*, *bnp*, *npra*, *npr1a*, *nprb*, *nprc*, *corin*, *furina* and *furinb*). Products were separated using capillary gel electrophoresis and expression quantified using the GenomeLab GeXP analysis system. Data were normalized to the housekeeping gene (*elfα*) and the mean displayed as fold change over basal expression found in untreated embryos. * *p* < 0.05, ** *p* < 0.01, *** *p* < 0.001 significantly different from control-treated embryos (*n* = 3 independent experiments).

## Data Availability

The data presented in this study are available on request from the corresponding author.

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
