# Peer review of "Pharmacological and Genetic Disruption of C-Type Natriuretic Peptide (nppcl) Expression in Zebrafish (Danio rerio) Causes Stunted Growth during Development"

_ijms, 2023, doi:10.3390/ijms241612921_

Round 1
Reviewer 1 Report
CNP encoding genes are associated with human diseases. Understanding the underlying mechanism is of great importance and clinical potential. In this manuscript, Andrew J. Lessey et al. identified zebrafish CNP orthologs and genes’ spatial temporal expression patterns using RT-qPCR and in situ hybridization. Then they performed functional analysis using exogenous CNP. Unexpectedly fish exhibited growth defects which likely resulted from reduced nppcl expression in a negative feed back manner. They confirmed this hypothesis using CRISPR/Cas9 targeting nppcl, which phenocopied CNP treatment. I appreciate authors’ effort to provide a whole picture of zebrafish CNPs. Data is very solid. Results are very interesting and will be useful to the field.
In the introduction, authors mentioned the rational of using zebrafish is to investigate the process of development, which they nicely demonstrated with multiple time points and developmental stages. To convince the audience zebrafish is indeed a great tool, authors might need to emphasize the advantages of using zebrafish with a couple more sentences.
Authors used CRISPR/Cas9 system to transiently knockdown expression of endogenous nppcl. According to the descriptions it looks like they were using a regular CRISPR/Cas9 approach with gRNA targeting coding sequence to generate indels. If so, it’s likely to generate mosaic knock-out after embryo injection. In addition, the compensatory induction of other close related genes is also a phenomenon when generating premature stop codon or truncated RNA. Such genetic adaptation is rare in knock-down approaches. And please provide gRNA sequences.
The exogenous CNP in this case may function dominant negatively. Is it possible to try any CNP of zebrafish origin? For example, purified nppcl protein, or nppcl transgenesis. It might be beyond the scope of this research but will be nice to address this question and make it really a great tool if able to show consistent results in the cases of transgenic mice and human treatment.
Fig1A is blurry. And highlight some key sequences mentioned in the text.
Fig2 lacks scale bar.
Line 220 should be ‘spatial’ expression.
Reviewer 2 Report
This manuscript presents an in-depth investigation of the role of C-type natriuretic peptide (nppcl) in the growth and development of Zebrafish. The study employs both pharmacological and genetic approaches to disrupt nppcl expression, focusing on the effects on embryonic development.
The research work begins by exploring the manipulation of CNP exposure during early development and the use of CRISPR/Cas9 to transiently knockdown expression of nppcl in developing zebrafish embryos. Key experiments include: (a) Injection of embryos with either nppcl gRNA and Cas9 RNA or Cas9 RNA alone, followed by morphometric analysis; (b) Utilization of multiplex RT-qPCR to assess the expression of various components of the natriuretic peptide system. The results reveal that CRISPR nppcl-treated embryos had significant growth impairments, including reduced head width, body length, and eye diameter. The study also identifies changes in the expression of other natriuretic peptides and highlights the potential compensatory mechanisms in response to nppcl inhibition.
Overall, this study unveils a novel developmental role for CNP in zebrafish, determining essential growth characteristics that have not been previously described in other models. The manuscript is methodologically rigorous and offers valuable insights into the regulation of growth in vertebrates, emphasizing the importance of intrinsic control of CNP for proper developmental growth. It makes a significant contribution to the field of developmental biology and offers a foundation for further research into the mechanisms governing growth and development.
Introduction section
1. Please take the time to define specialized scientific terms such as "achondroplasia" and "endochondral ossification." This will help ensure that the manuscript is accessible to readers who may not be specialists in this field. For example:
- Achondroplasia: A genetic disorder characterized by abnormal bone growth leading to dwarfism.
- Endochondral Ossification: A process by which cartilage is replaced by bone during development.
2. It's essential to emphasize the unique contributions of this study within the broader context of CNP research. Clearly outline what sets this work apart from previous studies, elaborating on the specific advances or insights that this research provides.
3. Given the predominant role of CNP in the central nervous system, provide a more comprehensive explanation of its functions and significance there. This could include details on how CNP interacts with other peptides or influences neural development.
4. The identification of nppcl as the zebrafish orthologue of human CNP is crucial to this study. Please provide a clear and detailed description of the specific methodologies, experimental designs, or statistical approaches used to make this identification. This will help readers understand the scientific foundation of your work.
5. Some of the sentences in the manuscript might be complex or lengthy, which can hinder understanding. Consider breaking them down into shorter, more straightforward statements. For example, instead of saying "Given the complexity of the mechanisms involved in endochondral ossification, this study aims to investigate the role of CNP in this process," you could say "Endochondral ossification involves complex mechanisms. This study aims to investigate CNP's role in this process."
Results
6. The manuscript could be enhanced by a more detailed description of the sequence homology and structural differences among CNPs. Consider including a table summarizing the sequence homology percentages and key structural differences, providing a clear overview for readers.
7. Although the temporal expression of CNPs during development is discussed, expanding on the biological implications of these findings would deepen the analysis. What might the different expression patterns signify in terms of development or function? Detailed explanations would provide more insights.
8. The manuscript should provide information on the statistical methods used, if any, to analyze the data. This includes statistical significance, error measurements, and the rationale for the chosen methods.
9. More context could be provided in terms of how this study fits within the existing literature. What novel insights does this study provide, and how does it expand on or contradict previous research?
10. A discussion of potential limitations or confounding factors in the study design, methodology, or interpretation of results would enhance the rigor of the manuscript.
11. A clear conclusion summarizing the main findings and their potential implications would round off the manuscript well. This could also include suggestions for future research directions. Additionally, the inclusion of potential limitations or challenges faced in the analysis would add to the rigor of the manuscript.
12. While the spatial expression of nppcl is described, the manuscript could be clearer in explaining the specific biological significance of its expression in regions like the forebrain, midbrain, and neuroendocrine regions.
13. Although similar temporal expression profiles are mentioned for nppc2, cnp3, and nppc4, a deeper comparative analysis of these genes with nppcl could elucidate their distinct roles.
14. The manuscript mentions receptors like npr1a and enzymes like furina and furinb. Including information on their functions and why they were selected for analysis would provide readers with essential context.
15. The manuscript could benefit from an extended discussion on how the observed temporal expression patterns correlate with the known developmental stages of zebrafish, and what these might imply about CNP function.
16. While the increase in TUNEL positive cells is noted, a more detailed explanation of how this links to growth inhibition might provide more insight.
17. If available, comparisons with similar studies in other organisms could contextualize the findings.
18. The section 2.5 could benefit from a table summarizing the fold-changes in gene expression for easy reference.
19. A discussion of why specific genes were affected differently by CNP treatment, in the context of known biological pathways, would add depth.
20. The sections 2.6 and 2.7 are clearly organized, detailing both morphometric changes and gene expression alterations due to CRISPR/Cas9 disruption of nppcl. However, some of the sentences are long and complex, which may impair readability.
21. Description of the selection 2.6 criteria for mild to moderate phenotypes should be expanded. Clarifying this would help others to reproduce the experiment.
22. A discussion on the biological significance of nppcl in zebrafish development and its connection to growth impairments would strengthen the section.
23. The detailed analysis of gene expression is commendable. However, a table or figure summarizing the changes in expression levels might make the data more accessible.
24. Add insight into why specific genes were affected by nppcl disruption, providing information on underlying mechanisms.
25. Treatment and CRISPR/Cas9 Disruption**: Provide a comparative analysis of the effects of exogenous CNP treatment versus CRISPR/Cas9 disruption of nppcl. This comparison would offer a comprehensive understanding of the subject matter.
Discussion
The section is comprehensive and provides a good summary of the findings, linking them to existing knowledge. However, the lengthy paragraphs may be challenging for readers, and the flow of ideas could be better organized.
26. The discussion on the role of CNP in development is detailed but could benefit from a more succinct summary. The authors should highlight key takeaways and unresolved questions.
27. While there is a good comparison with mammals, including other organisms and relating the findings to broader vertebrate evolution could enhance the discussion.
28. The discussion on compensatory mechanisms following CRISPR nppcl treatment is intriguing but needs further elaboration and substantiation with evidence from literature.
29. The underlying mechanisms for the observed effects are mentioned, including feedback regulation and receptor desensitization. However, these need to be elaborated more clearly, and hypotheses should be explicitly stated.
30. Provide clearer explanations of the biological mechanisms and make explicit hypotheses or predictions.
31. Include a more extensive comparison with other organisms to place the findings in a broader evolutionary context.
32. Discuss the potential implications of the findings, both in terms of understanding developmental biology and possible therapeutic applications.
Other
33. A title should succinctly encapsulate the main focus and findings of the manuscript. Here are two suggestions that might better fit the content of the manuscript. "Mechanistic Insights into the Effects of nppcl Disruption on Gene Expression in Zebrafish Development" or "Exploring the Impact of nppcl Disruption on Specific Genes: Unraveling the Underlying Mechanisms".
The language used in the manuscript is clear and well-structured, fitting for an academic setting. However, some parts of the text contain complex or lengthy sentences that may impede understanding. To enhance readability, it might be beneficial to break these sentences into shorter, more concise statements. For specific examples and further suggestions, please refer to the Reviewer's report.
Round 2
Reviewer 2 Report
The author has addressed all of my concerns, and I have no further suggestions. The manuscript is suitable for publication in IJMS.